# Rebuilding Tree Cover in Deforested Cocoa Landscapes in Côte d'Ivoire: Factors Affecting the Choice of Species Planted

Alain R. Atangana *, Juvenal Zahoui Gnangoh, Allegra Kouassi Yao, Thomas d'Aquin Kouakou, Anatole Mian Ndri Nda and Christophe Kouamé

ICRAF West and Central Africa Region, Angré 7ème Tranche, Cocody, Abidjan 08 BP 2823, Côte d'Ivoire; J.Zahoui@cgiar.org (J.Z.G.); A.Kouassi@cgiar.org (A.K.Y.); T.Kouakou@cgiar.org (T.d.K.); A.Mian@cgiar.org (A.M.N.N.); C.Kouame@cgiar.org (C.K.)
* Correspondence: A.Atangana@cgiar.org; Tel.: +225-22504839

**Abstract:** Intensive cocoa production in Côte d'Ivoire, the world's leading cocoa producer, has grown at the expense of forest cover. To reverse this trend, the country has adopted a "zero deforestation" agricultural policy and committed to rehabilitating its forest cover through the planting of high-value tree species in cocoa landscapes using a participatory approach. However, little is known regarding the factors influencing farmers' introduction of high-value tree species to cocoa landscapes. We tested the hypothesis that ten previously reported factors to influence agroforestry system adoption to predict the number and choice of tree species that farmers introduce to cocoa farms. We interviewed 683 households in the cocoa-producing zone of Côte d'Ivoire and counted tree species on their cocoa farms. On average, two tree species were recorded per surveyed farm. Generalized Poisson regression models revealed that, in the cocoa production area, experience in tree planting and expected benefits, including income and food, influence tree species introduction through planting or "retention" when clearing land for cocoa establishment. The age of the farmer also influenced ($p = 0.017$) farmers' tree species planting on cocoa farms. Fewer tree species were introduced into current intensive cocoa-production areas than in "old cocoa-loops" and forested areas. The number of tree species introduced to cocoa farms increased with expected benefits and experience in tree planting. The number of planted tree species also increased with farmers' age. Tree species were mostly selected for the provision of shade to cocoa, production of useful tree products (38%), and income from the sale of these products (7%). Fruit tree species were the most planted, while timber tree species were mostly spared when clearing land for cocoa production.

**Keywords:** biodiverse tree-based systems; experience in tree planting; farmer-driven reforestation; tree benefits; trees on farms





## 1. Introduction

Land conversion to intensive cash crop production has often resulted in the rapid disappearance of forests in the tropics, as seen in Côte d'Ivoire in 2018/2019 when it was the top producer of cocoa, contributing over 45% of global production [1]. Cocoa is currently produced by more than 800,000 smallholder farmers, contributing over 22% of the gross domestic product of the country. The boom in cocoa production since the 1960s has been supported by rising global demand and price incentives. This has resulted in increased demand for land for higher-yielding cocoa hybrids grown as monocrops in full sun [2,3] between 1970 and 1975. Indeed, since the 1960s, forest cover nationwide has declined by 80% (13 million hectares) [4]. To reverse deforestation and restore tree cover, the country has launched a "zero deforestation objective"—farming without destroying forests. Additionally, as its contribution to the Bonn Challenge through the AFR100 initiative [5], the country has pledged to restore five million hectares of degraded land and to reconstitute three million ha of the forest by 2030. One priority intervention of this initiative is to enrich agricultural landscapes by planting high-value trees through a new forest code (Law No

2019-675, dated 23 July 2019) and the subsequent decree number 2019-818 promoting the creation of agroforests in the country.

Tree planting on farms or degraded land is recognized as an efficient strategy to improve human welfare through the provision of edible fruits, nuts, leaves and fuelwood, and cash from the sale of these products, while also sequestering carbon to mitigate climate change [6–9]. However, to minimize undesirable outcomes, such as increased social inequity and invasion by alien species [10], the design and implementation of tree planting programs should be carefully considered. Indeed, to be truly successful, tree planting for economic, social, ecological, or esthetic reasons should consider the objectives and needs of local farmers in the target areas. For example, the worldwide tree domestication program that has been carried out in the tropics since the 1990s is farmer-driven and market-oriented [9,11]. This participatory tree domestication has proven efficient for diversifying food production while generating income for farmers and providing multiple social, economic and environmental benefits [7,9], based on farmers' priority ranking of tree species to be introduced on farms in West and Central Africa [12], Southern Africa [13], Oceania [14,15], Southeast Asia [16], and Latin America [17]. Indeed, farmers often protect, retain, or plant useful, "socially important", and/or marketable tree species [18] when clearing forest or fallow patches for farming and cocoa production. Interestingly, the species planted often vary between sites; for instance, bush mango trees (*Irvingia wombolu/I. gabonensis*) in Nigeria versus safou (*Dacryodes edulis* H.J. Lam) in Cameroon [19]. In contrast, trees are seldom found on cocoa farms in Côte d'Ivoire and Ghana [20,21], where less is known about the farmers' interest in trees on cocoa farms. Thus, it is encouraging that in Côte d'Ivoire, the Coffee-Cocoa Board has recently indicated that, using a participatory approach [22], farmers' opinions and objectives should now be taken into account when selecting trees for planting on cocoa farms [23].

The factors that are reputed to influence the adoption of agroforestry systems include the socioeconomic characteristics of the farmer (i.e., potential adopter: age, gender, education level, cultural and eating habits, on-farm income, food security, perceptions and attitudes towards trees, marketing of products, local knowledge, and well-being), the external environment of the adopter (land tenure and tree rights, market access), and factors related to new technology (labor requirements, investment cost, long-term nature of the investment, and expected benefits [24–29]). However, the relative contribution of these factors to the decision to adopt may vary with practice and context [25,27–32]. For example, social network effects, ethnic group, and geographic zone were found to determine the presence of trees and their density on cocoa farms in the Soubré region in Côte d'Ivoire [20], as did the severity of cocoa diseases and the existence of programs for extension and certification of cocoa, However, in the humid forest zones of Cameroon and Nigeria, strategies for growing fruit trees were identified as suitable for market access, land-use, and access to forest resources [33]. In the Eastern region of Ghana, farmer interviews found that *Spathodea campanulate* P. Beauv. (Bignoniaceae), *Terminalia superba* Engl. and Diels. (Combretaceae), and *Terminalia ivorensis* A. Chev. (Combretaceae) [21] were the most popular tree species. Inventories of cocoa farms in the Baoulé region of Côte d'Ivoire have recorded tree numbers [34], but the factors related to them remain unknown.

In Côte d'Ivoire, local and foreign migrants are currently interested in purchasing or renting farmland in the current "cocoa-loop" (the west, center, and south of the country), while historically it was the Eastern side of the country that was first favored for cocoa production [35]. Three categories of cocoa farmers who have impacted forests and old fallows by buying or renting land can therefore be identified: (i) the autochthonous, who are typically landowners, (ii) the allochthonous, who are local migrants from within Côte d'Ivoire, and (iii) the allogenous, who are foreign migrants. Hence, the migration status of the farmers probably affects land tenure status (land certificate or title) based on customary rights. Further, both the allochthonous and allogenous are often reported to invade state "classified forests", regulated by laws defining limits of use and timber production, as well as national parks, for farming. Consequently, in Côte d'Ivoire, the migration status of a

farmer needs to be included among the factors likely to influence tree planting on cocoa farms.

Rebuilding forest cover in Côte d'Ivoire is related to the restoration of biodiversity, especially as the forest zone is an extension of the "Upper Guinea center of plant species diversity" [36], with the Taï Forest being a center of endemism [37] due to a high level of observed plant species richness and 2.6% endemism (Sosef et al. 2017) [38]. These forests contain many high-value timber species, including *Cedrela odorata* L. (Meliaceae), *Entandrophragma angolense* (Welw.) C. DC. (Meliaceae), *Khaya ivorensis* A. Chev. (Meliaceae), *Mansonia altissima* (A. Chev.) A. Chev. (Malvaceae), *Milicia excelsa* (Welw.) C. C. Berg (Moraceae), *Nesogordonia papaverifera* (A. Chev.) Capuron (Malvaceae), *Terminalia ivorensis,* and *Triplochiton scleroxylon* K. Schum (Malvaceae). However, deforestation has destroyed the habitats of these species, as well as those of terrestrial fauna. To address these issues, agroforestry offers a valuable alternative for conserving forest biodiversity and sustaining ecosystem services within agricultural landscapes [39]. For example, in the humid forest zone of Cameroon [40], multi-strata cocoa agroforestry systems have been reported to contain *Musa* species, *Elaeis guineensis* Jacq. (Arecaceae), *Dacryodes edulis* H.J. Lam (Burseraceae), *Persea americana* Mill. (Lauraceae), *Mangifera indica* L. (Anacardiaceae), *Funtumia elastica* (Preuss) Stapf (Apocynaceae), *Albizia adianthifolia* (Schumach.) W.F. Wight (Fabaceae), *Terminalia superba*, *Ficus exasperata* Vahl (Ficeae), *Pycnanthus angolensis* (Welw.) Warb. (Myristicaceae), *Margaritaria discoidea* (Baill.) G.L. Webster (Phyllanthaceae), *Ficus mucuso* Welw. Ex Ficalho (Moraceae), and *Triplochiton scleroxylon*. These cocoa agroforests also enhance the abundance and diversity of soil fauna [41,42]. The agroecological importance of such diversity in cocoa and coffee agroforests, in terms of pest and disease control, has been confirmed [43].

As a starting point for the development of biodiverse and multi-strata cocoa-based agroforests, the present study aims to understand the factors affecting the tree species introduced into, or retained in, cocoa farms by farmers. The objective of the study is to determine factors influencing the number of tree species associated with planting or "retention" during the clearance of forest or fallow patches for cocoa cultivation. We hypothesize that the factors previously reported influencing the adoption of agroforestry systems to predict the number of tree species that farmers introduce into cocoa farms.

## 2. Materials and Methods

### 2.1. Sites of Study

Sixteen study sites were sampled in the cocoa-producing zone of Côte d'Ivoire, which stretches geographically from east to west (Figure 1) following the shifting "cocoa-loop" from the east (Indénié-Djuablin and South-Comoé) to the center (Lôh-Djiboua, Marahoué) and west (Guémon; Gbôklé, Cavally and Haut-Sassandra; Figure 1) regions. The sites correspond to the intensity of cocoa production in the country, as most cocoa production currently occurs in the south and center, and the remaining forest patches are found in the west. It is worth noting that these forests are currently being invaded by farmers in search of land for cocoa cultivation.

The study sites were in the Guinean and the Sudano-Guinean zones of Africa. The Guinean zone of Côte d'Ivoire was formerly covered by evergreen and semi-deciduous dense humid forests; the zone is characterized by a sub-equatorial climate with two rainy and two dry seasons. The average annual rainfall ranges from 1400 to 2500 mm [44]. The annual air relative humidity and annual mean temperature in the area range between 80 and 90%, and between 25 and 33 °C, respectively [44]. The Sudano-Guinean zone is found between the Guinean zone in the southern part of the country and the Sudanese zone in the north. Soils of both the Guinean and Sudano-Guinean zones are mostly of the desaturated ferralitic type, on a Precambrian base [44].

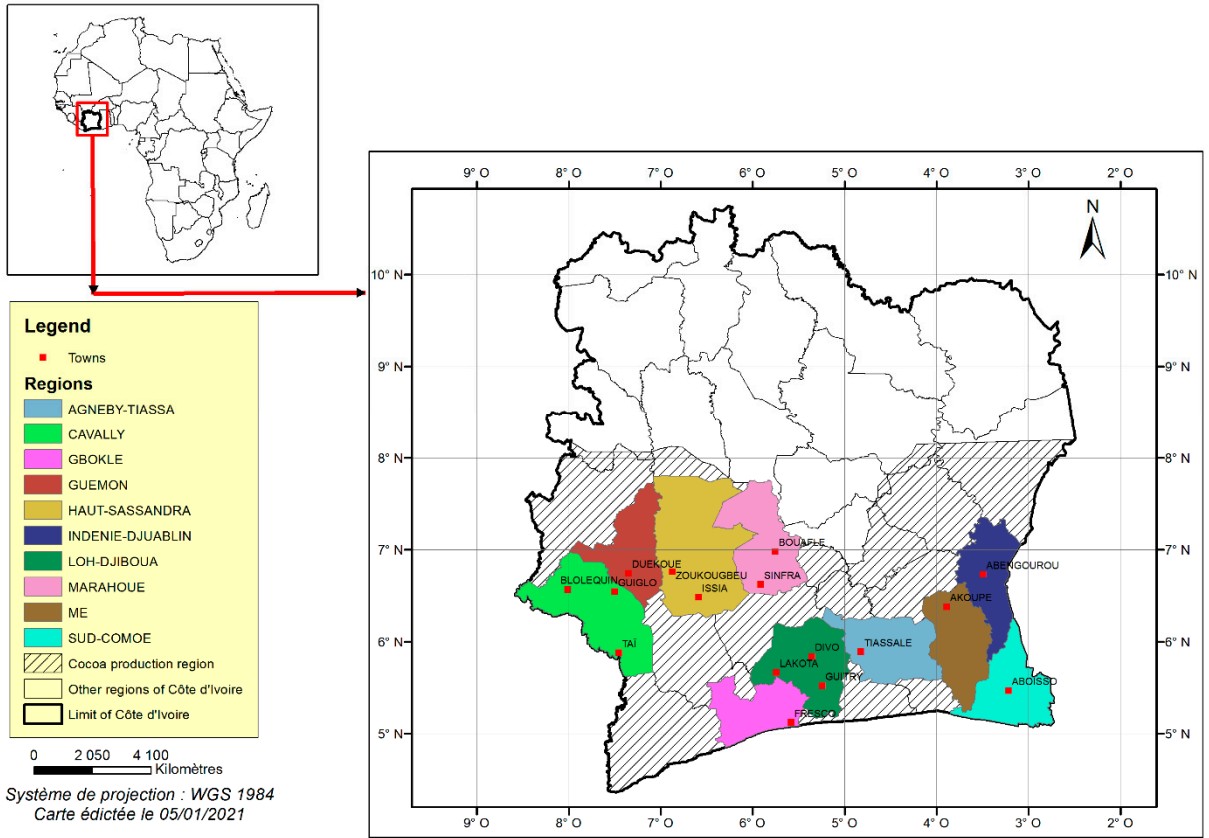

**Figure 1.** Sampling sites of a study aimed at identifying factors that drive tree species introduction into cocoa farms by farmers in Côte d'Ivoire.

### 2.2. Sampling

A total of 683 households (Table 1) were interviewed, and the number of trees on their cocoa farms were counted in this study. Each village is subdivided into units or sections by the National Institute of Statistics of Côte d'Ivoire to ease the surveys. Systematic sampling [45] was used to choose the households to be interviewed. The sampling step was calculated as follows [45]:

Step = total number of producers in the section/number of producers to be surveyed per section.

The step obtained allowed the sampling of the first household in the section. The second household to be interviewed in the section was drawn as follows:

Second household = step + (total number of producers in the section/total number of producers to be surveyed in the section).

Then all sections are covered in the sampling site using this method.

### 2.3. Description of the Questionnaire

The questionnaire used included twenty-nine questions—both closed format questions and single/multiple-choice questions—as well as exploratory questions allowing several responses. For example, we allowed respondents to list all reasons for introducing trees to their cocoa farms, as well as the species they planted or spared when clearing land for cocoa planting. Each interview lasted about forty-five minutes.

**Table 1.** Number of surveyed households and localization of sampling sites to determine the drivers of tree introduction to cocoa farms in Côte d'Ivoire.

| Geographical Zone | Region | Number of Surveyed Households |
|---|---|---|
| East | Indénié-Djuablin | 64 |
| | Sud-Comé | 152 |
| Center | Lôh-Djiboua | 89 |
| | Marahoué | 56 |
| South | Mé | 36 |
| | Agneby-Tiassalé | 26 |
| West | Cavally | 98 |
| | Gbôklé | 26 |
| | Guémon | 67 |
| | Haut-Sassandra | 69 |
| Total | | 683 |

*2.4. Data Collection and Analysis*

Before the start of the survey, we obtained the authorization of the telecommunications regulatory authority in Cote d'Ivoire, which handles any surveys using personal data. The consent of each interviewee was requested before starting the interview. Data were collected in November and December 2016. Interviews were carried out using the Census and Survey Processing System (CSPro) software [46]. This software is a suite of tools for collecting data using electronic devices (e.g., personal digital assistants and Android smartphones). The collected data were stored in the memory of the electronic device used and retrieved by connecting the device to a computer.

Data analysis was performed using the generalized Poisson regression model for over-dispersed and under-dispersed count data [47,48]. The number of tree species introduced (i.e., planted or retained during land clearing for cocoa planting) on cocoa farms was considered as a response variable, while the covariates consisted of cocoa-sourcing areas (east, center, south and west), gender, respondent's age, an education level (none, primary, secondary, university), migration status (i.e., autochthonous, allochthonous, allogenous), matrimonial status (i.e., single, married, divorced, widowed), farm size, land tenure status, experience in tree planting, and expected benefit from the tree species (i.e., no benefits, one benefit, multiple benefits).

**3. Results**

The generalized Poisson regression model converged and was very highly significant ($p < 0.0001$) for both the number of tree species planted and the number retained during land clearing for cocoa establishment. On average, $2 \pm 1.10$ tree species were recorded per cocoa farm, and the forest was the most common previous land-use type (66.2%), followed by fallows (21.3%).

*3.1. Tree Species Planting*

Tree planting experience and its expected benefits had very highly significant effects on the number of tree species that farmers planted on cocoa farms (Table 2). The number of tree species planted increased significantly ($p = 0.0001$) with expected benefits, as on average, three tree species were planted when farmers expected multiple benefits, whereas only 2.4 and 1 tree species were planted when the farmer expected, respectively, one or no benefits (Table 2). The expected benefits from trees were mostly the provision of shade and production of edible fruits and nuts (Figure 2). The most frequently planted tree species were *Persea americana*, *Mangifera indica*, *Cola nitida* Schott and Endl; (Malvaceae), *Citrus sinensis* L. Osbeck (Rutaceae), *Terminalia superba*, *Milicia excelsa*, *Ricinodendron heudelotii* (Baill.) Heckel (Euphorbiaceae), and *Triplochiton scleroxylon* (Figure 3). It is worth noting

that avocado, mango, and orange trees are exotic fruit tree species. Of the indigenous tree species, only Cola and *R. heudelotii* produce edible nuts.

**Table 2.** Generalized Poisson regression model estimates of the factors driving tree species planting on cocoa farms in Côte d'Ivoire.

| Source of Variation | Coefficient | z | *p*-Value | Incidence Rate Ratio | Confidence Interval |
|---|---|---|---|---|---|
| Sociodemographic characteristics | | | | | |
| Cocoa sourcing zone | | | | | |
| East | 0 | | | 1 | |
| West | −0.0272 | −0.57 | 0.567 | 0.9731779 | 0.8866995–1.06809 |
| South | −0.219 ** | −2.73 | 0.006 | 0.8032214 | 0.6863705–0.9399657 |
| Center | −0.192 *** | −3.44 | 0.001 | 0.8255385 | 0.7401966–0.92072 |
| Age (years) | 0.00370 * | 2.38 | 0.017 | 1.003704 | 1.000648–1.006769 |
| Gender | | | | | |
| Male | 0 | | | 1 | |
| Female | −0.132 | −1.12 | 0.265 | 0.8760019 | 0.6941601–1.105479 |
| Household size | 0.00539 | 0.94 | 0.349 | 1.0054 | 0.9941388–1.016788 |
| Matrimonial status | | | | | |
| Married | 0 | | | 1 | |
| Single | 0.0614 | 0.75 | 0.451 | 1.063301 | 0.9065681–1.24713 |
| Widowed | −0.179 | −0.84 | 0.398 | 0.8364612 | 0.5526838–1.265945 |
| Divorced | −0.0596 | −0.35 | 0.729 | 0.9421713 | 0.6728606–1.319273 |
| Education level | | | | | |
| No | 0 | | | 1 | |
| Primary | −0.0641 | −1.31 | 0.192 | 0.9378716 | 0.851798–1.032643 |
| Secondary | 0.0176 | 0.28 | 0.777 | 1.017782 | 0.9009709–1.149737 |
| University | 0.0242 | 0.11 | 0.914 | 1.02452 | 0.6597819–1.590891 |
| Migration status | | | | | |
| Allochthonous | 0 | | | 1 | |
| Autochthonous | −0.0588 | −1.00 | 0.318 | 0.9429335 | 0.8401908–1.05824 |
| Allogenous | −0.0183 | −0.35 | 0.729 | 0.9818601 | 0.8851158–1.089179 |
| Farm characteristics | | | | | |
| Land tenure status | | | | | |
| Owner of the land | 0 | | | 1 | |
| Tenant of the land | −0.0511 | −0.74 | 0.462 | 0.9501761 | 0.8293153–1.088651 |
| Farm size (ha) | 0.00147 | 0.44 | 0.661 | 1.00147 | 0.9949089–1.008074 |
| Knowledge of trees | | | | | |
| Experience in tree planting | 0.0157 *** | 7.08 | 0 | 1.01583 | 1.011421–1.020258 |
| Expected benefit | | | | | |
| No use | 0 | | | 1 | |
| Unique utility | 0.897 *** | 3.93 | 0 | 2.45109 | 1.566631–3.83488 |
| Multiple utility | 1.008 *** | 4.26 | 0 | 2.739624 | 1.722761–4.356693 |

* *p* < 0.05: significant, ** *p* < 0.01: highly significant *** *p* < 0.001: very highly significant.

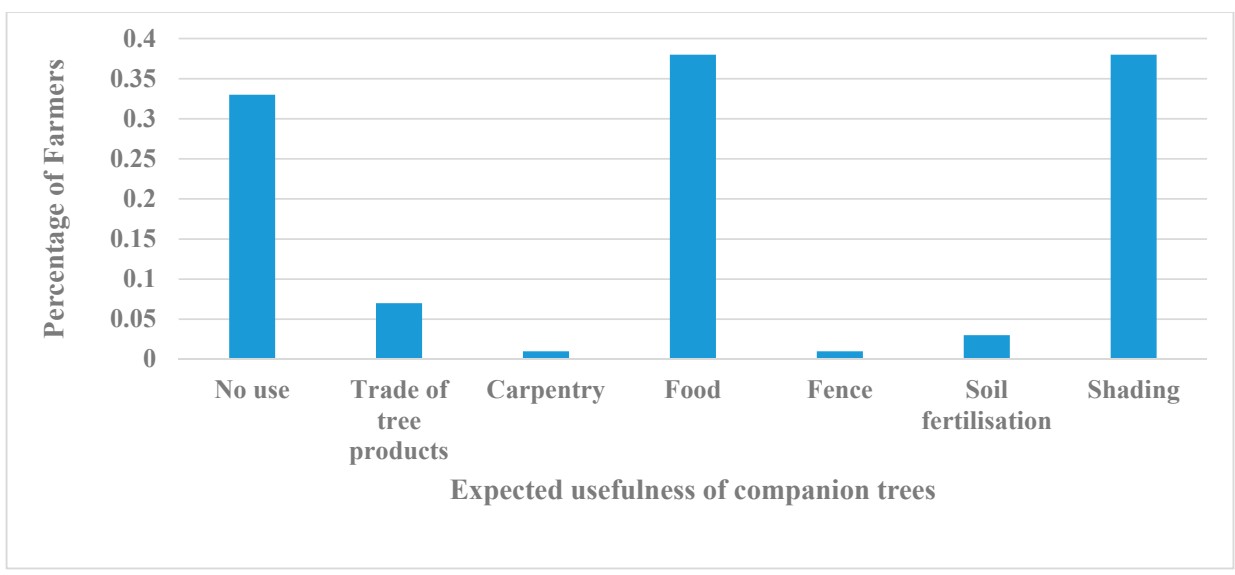

**Figure 2.** Farmers' expected benefits from planted trees on cocoa farms in Côte d'Ivoire.

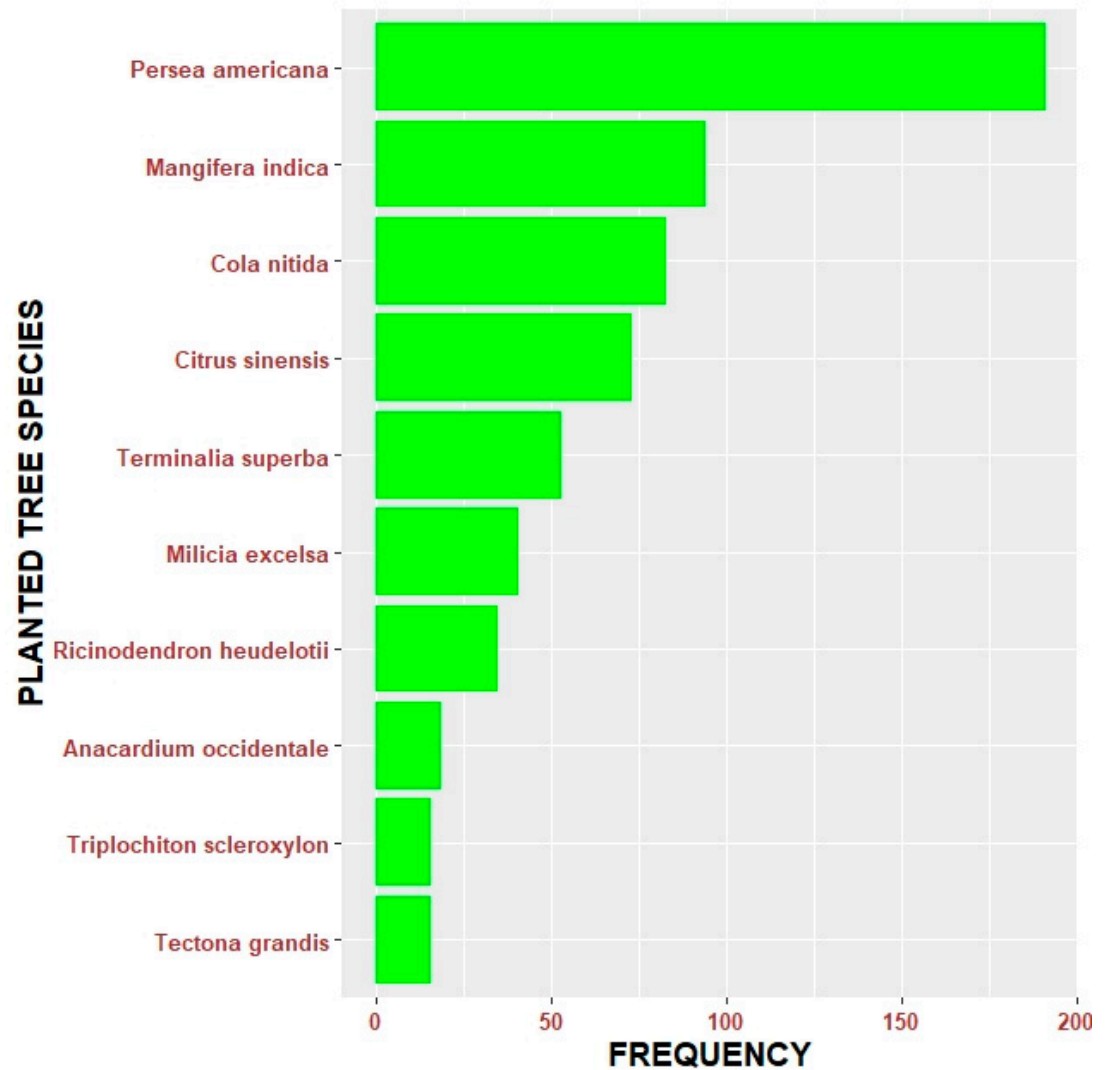

**Figure 3.** Tree species most frequently planted by farmers on cocoa farms in Côte d'Ivoire.

The location of the cocoa production areas significantly affected the number of tree species that farmers planted on cocoa farms (Table 2), with the least being planted in the center, south, and west, and the most in the east (Table 2). The age of the farmers significantly affected the number of species planted as more trees were planted by older farmers ($p$ = 0.017; Table 2), while gender, household or family size, matrimonial status, farm size, education level, migration status, farm size, and land tenure status had no effect on the number of tree species planted (Table 2).

### 3.2. Tree Species Retained on Land Cleared for Cocoa Farming

The number of trees retained was highly significantly ($p < 0.001$) affected by tree planting experience, while production areas and expected benefits also significantly affected this variable (Table 3). The number of tree species retained when clearing land for cocoa planting increased with benefits and decreased in intensive cocoa-production areas (Table 3). The species that were most retained during land clearance for cocoa farms included timber and indigenous and exotic fruit and nut species, namely *Milicia excelsa*, *Ceiba pentandra* (L.) Gaertn. (Malvaceae), *Terminalia superba*, *Triplochiton scleroxylon*, *Ricinodendron heudelotii*, *Cola nitida*, *Adansonia digitata* L. (Malvaceae), *Khaya ivorensis* A. Chev. (Meliaceae), *Terminalia ivorensis*, *Persea americana*, *Mangifera indica*, *Tieghemella heckelii* (A. Chev.) Pierre ex Dubard (Sapotaceae), *Irvingia wombolu,* and *Elaeis guineensis* (Figure 4).

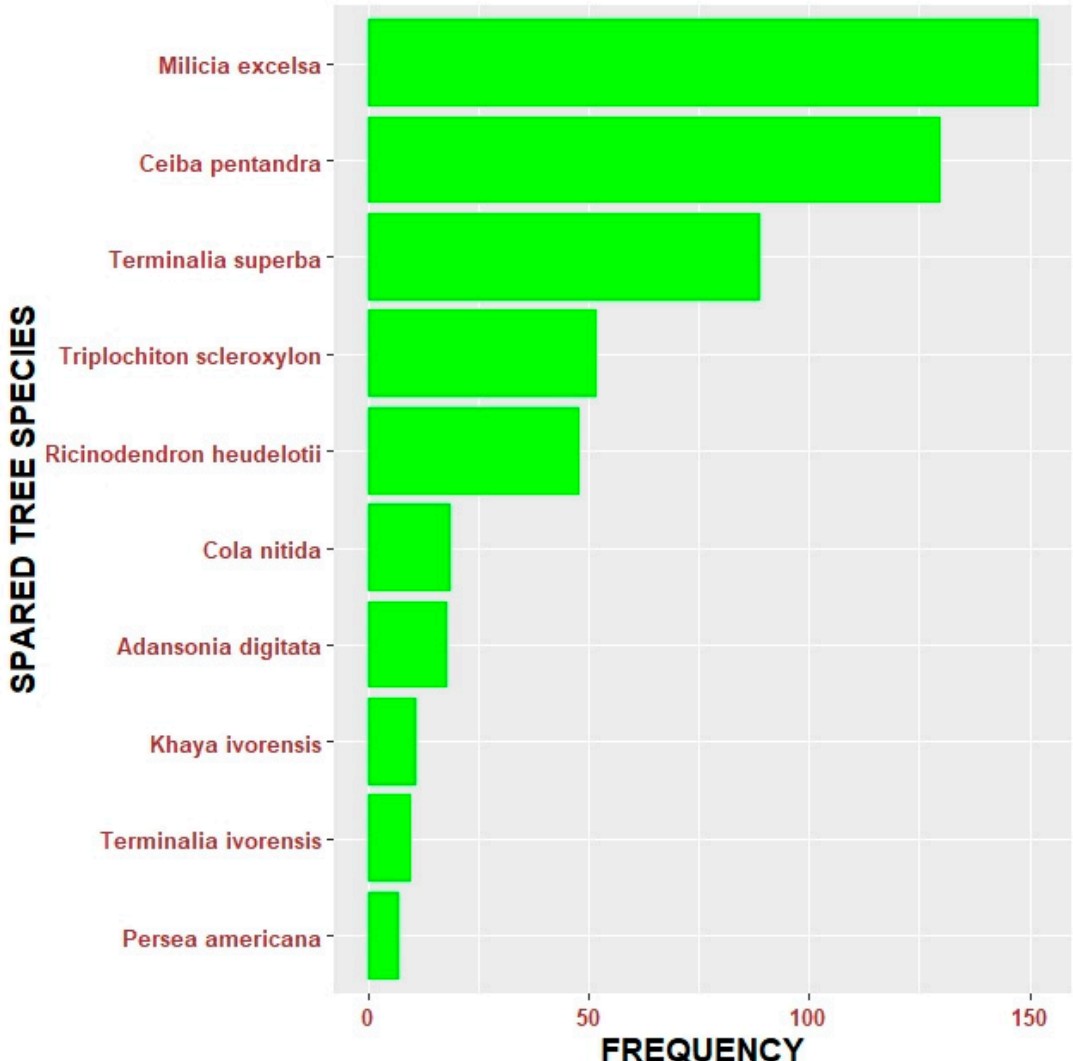

**Figure 4.** Tree species that are most frequently spared when clearing land for cocoa farming in Côte d'Ivoire.

**Table 3.** Generalized Poisson regression model estimates of the factors driving retention of tree species when clearing land for cocoa farming in Côte d'Ivoire.

| Source of Variation | Coefficient | z | *p*-Value | Incidence Rate Ratio | Confidence Interval |
|---|---|---|---|---|---|
| Sociodemographic characteristics | | | | | |
| Cocoa sourcing zone | | | | | |
| East | 0 | | | 1 | |
| West | −0.0342764 | −0.51 | 0.609 | 0.9663044 | −0.165711–0.09716 |
| South | −0.2379414 * | −2.4 | 0.041 | 0.7882489 | −0.4662497–0.00962 |
| Center | −0.4079864 *** | −4.54 | 0 | 0.6649879 | −0.58405–0.231922 |
| Age (years) | 0.0031283 | 1.41 | 0.159 | 1.003133 | −0.001227–0.007484 |
| Gender | | | | | |
| Male | 0 | | | 1 | |
| Female | −0.1258805 | 0.75 | 0.456 | 0.8817202 | −0.4569015–0.20514 |
| Household size | 0.0049735 | 0.59 | 0.558 | 1.004986 | −0.011659–0.021606 |
| Matrimonial status | | | | | |
| Married | 0 | | | 1 | |
| Single | 0.10947 | 0.9 | 0.367 | 1.115689 | −0.1284096–0.347353 |
| Widowed | −0.003539 | −0.01 | 0.99 | 0.9964672 | −0.539174–0.532096 |
| Divorced | −0.0095684 | −0.04 | 0.968 | 0.9904772 | −0.481664–0.462527 |
| Education level | | | | | |
| No | 0 | | | 1 | |
| Primary | −0.0982293 | −1.35 | 0.176 | 0.906441 | −0.240433–0.043975 |
| Secondary | −0.0426565 | −0.46 | 0.645 | 0.9582404 | −0.224025–0.138711 |
| University | 0.0285906 | 0.08 | 0.936 | 1.029003 | −0.667276–0.724457 |
| Migration status | | | | | |
| Allochthonous | 0 | | | 1 | |
| Autochthonous | −0.0599897 | −0.68 | 0.498 | 0.9417743 | −0.233578–0.113599 |
| Allogenous | 0.0023149 | −0.03 | 0.976 | 1.002318 | −0.150771–0.155401 |
| Farm characteristics | | | | | |
| Land tenure status | | | | | |
| Owner of the land | 0 | | | 1 | |
| Tenant of the land | −0.0621578 | −0.62 | 0.535 | 0.9397346 | −0.258641–0.134325 |
| Farm size (ha) | −0.0028882 | −0.51 | 0.608 | 0.9971159 | −0.013932–0.008156 |
| Knowledge of trees | | | | | |
| Experience in tree planting | 0.0298506 *** | 10.5 | 0 | 1.030301 | 0.0242782–0.035423 |
| Expected benefit | | | | | |
| No use | 0 | | | 1 | |
| Unique utility | 0.3661164 | 1.58 | 0.115 | 1.442123 | −0.088621–0.820854 |
| Multiple utility | 0.5656596 * | 2.29 | 0.022 | 1.760609 | 0.0804973–1.050822 |

\* *p* < 0.05: significant, \*\*\* *p* < 0.001: very highly significant.

We found no significant difference between sites previously under different land-use systems, although 50% of surveyed households introduced more tree species in plots that had been previously used for food cropping (Table 4).

**Table 4.** Influence of previous land-use systems on the number of tree species that are introduced to cocoa farms in the cocoa-producing zone of Côte d'Ivoire.

| Source of Variation | Frequency | Average | 50% of Farms | 25% of Farms | 75% of Farms | Standard Deviation | Standard Error (Mean) |
|---|---|---|---|---|---|---|---|
| Fallow | 150 | 1.96667 | 2 | 1 | 3 | 1.070763 | 0.0874274 |
| Food crops | 9 | 2.4444 | 3 | 1 | 3 | 1.236033 | 0.412011 |
| Perennial crops | 66 | 1.77273 | 1 | 1 | 3 | 0.957488 | 0.1178586 |
| Forest | 458 | 2.1092 | 2 | 1 | 3 | 1.12709 | 0.0526654 |
| | 683 | 2.04978 | 2 | 1 | 3 | 1.104714 | 0.0422707 |

## 4. Discussion

Studies on the decisions affecting the adoption of agroforestry have most often focused on the identification of factors influencing farmers' choice to acquire and use agroforestry technologies, while the factors influencing the biodiversity of agroforestry systems have generally been overlooked by science. The present study is innovative in that we determined the number of tree species that farmers plant on cocoa farms from the socioeconomic characteristics of their households, their objectives and the surrounding environment, and from farm characteristics. The results of this study will help identify actions to develop biodiverse and multi-strata cocoa agroforestry using a participatory approach. The results of the present study are in agreement with that of Smith Dumont et al. [49], who reported that smallholder farmers in Côte d'Ivoire plant or retain only $4 \pm 1.8$ tree species per cocoa farm in the Bas-Sassandra region (southwest of Côte d'Ivoire). This low tree species diversity in cocoa farms has also been recorded in other top producing countries like Ghana (2.4 to $5.10 \pm 0.38$ tree species per farm) [50,51] and Indonesia (two legume tree species per cocoa farm) [52]. These results are in contrast with those of [40] in the humid forest zone of Cameroon, where cocoa is grown under shade, and where as many as 20 tree species have been reported. Such results indicate that cocoa agroforests in West Africa are not very biodiverse compared with those in Central Africa. As biodiversity is very important for ecological stability, it seems that efforts need to be taken to convince cocoa farmers of the importance of diversification in top-producing cocoa bean countries. A valuable strategy would be to identify farmers' expected benefits from trees, identify the species that provide such benefits, and then design cocoa agroforestry practices that include the most relevant and important trees. In addition to this, there is the opportunity to design agroforestry practices that could also maximize food security and social and economic benefits from the choice of tree, as advocated by Leakey [7,9].

The results of the present study indicate that farmers' tree planting activities on cocoa farms depend on the expected benefits (i.e., motivation) and ability (i.e., experience in tree planting and farmers' age). In the Sudanian Savanna Zone of Burkina Faso, such tree planting decisions in parkland farming systems were also found to be influenced by silvicultural knowledge and perceived economic benefits [31]. Elsewhere, such as in the Philippines, reforestation incentives and the dependence of local people on forests were listed among the drivers of reforestation projects [53]. The results of the present study are also congruent with the farmer-driven and market-oriented strategy of high-value tree species integration into croplands, as implemented by the World Agroforestry (ICRAF) in the tropics [7,54]. Further, though the provision of shade, consumption, and trade of agroforestry products ranked top among farmers' expected benefits for tree species introduced into cocoa farms, the number of species to be planted was found to depend on the multiplicity of benefits.

Experience in tree planting benefits from the practice. The results of the present study highlight the importance of farmers' continuous training in agroforestry, as undertaken in rural resource centers (RRCs) supported by World Agroforestry in Cameroon [55]. RRCs

are aimed at training and "updating former trainees" in agroforestry; these centers are equipped with a nursery, a stock plant garden for vegetative propagation, and the library necessary for farmers' training. For this reason, replicating agroforestry RRCs in the cocoa-production zone of Côte d'Ivoire is recommended, so that farmers can acquire and perform the necessary skills of tree domestication and planting in agroforestry systems. Other approaches to training, such as "champ-école" (i.e., school farms), are being tested by non-governmental and governmental organizations involved in training farmers in agroforestry.

In Côte d'Ivoire, the areas that are suitable for cocoa production are aggregated into "cocoa-loops", with the cultivation in the old loops involving cocoa under shade trees, while in the current loop, unshaded cocoa monocropping prevails. The latter is thought to be more intensive for farmers seeking higher cocoa yields [56]. This explains the results of the current study, in which lower numbers of tree species per farm were recorded in the current loop. However, there are numerous concerns regarding the unsustainability of high yields from unshaded monocropped cocoa, especially due to the ecological collapse of cocoa production in the Atlantic Coast area of Brazil, which resulted in witches' broom disease and an economic crisis [57,58]. One solution that has been proposed is the establishment of multi-strata cocoa agroforests for greater ecological stability [43,59,60]. This more holistic approach includes the adoption of ecological science, farmers' objectives, and appropriate policies to sustain agricultural production [8] based on the principle of "land-maxing" [7]. We, therefore, recommend that cocoa-based agroforestry projects in Côte d'Ivoire should seek to optimize the integration of trees within cocoa farms by optimizing both the number of trees and the products and services provided by different tree species in ways that maximize economically, agronomic (i.e., cocoa yield), and environmental (i.e., sustainable production) benefits. Further studies are therefore needed to determine how tree diversity increases yield and the overall production in cocoa systems through (i) the management of soil fertility and ecological health [61–64] and (ii) the production of useful and marketable products from a range of different tree species [7].

The design of agroforestry systems in Côte d'Ivoire to maximize overall production and benefits from cocoa production could be based on the results of the present study, which indicate that farmers retain and plant both indigenous and exotic species for producing timber and a range of non-timber forest products, in congruence with their objectives and priorities for cocoa agroforestry. Exotic fruit tree species would help to reduce food and nutritional insecurity in cocoa-producing zones. This is important, as levels of food insecurity in the current cocoa-loop of the Nawa region reached 49% in 2014 and 39.8% in 2015 for children under five years among the 1069 households that were surveyed [65,66]. Similar levels of malnutrition have been reported in cocoa-producing zones in Ghana [67], indicating the need to diversify and increase food production by introducing agroforestry trees into cocoa farms. To assist this, there is an urgent need to develop a database of agroforestry tree species in Côte d'Ivoire. This database should include trees that produce agroforestry products, such as edible and marketable fruits and nuts, as well as exotic fruit species, timber and medicinal trees, and nitrogen-fixing fertilizer trees that boost food security from staple food crops. The database should also consider tree species that occupy different strata for shade provision to cocoa.

## 5. Conclusions

The present study clearly revealed that the expected benefits and experience in relation to tree planting drive farmers' tree planting activities on cocoa farms in Cote d'Ivoire. Progress towards the development and deployment of biodiverse multi-strata cocoa agroforestry in the country requires the establishment of a species database that details the benefits that farmers can expect from trees, including edible and marketable products. Likewise, there is a need to implement and upscale continuous training centers for farmers to allow them to acquire and perfect agroforestry skills. This would increase farmers' experience in tree planting and facilitate the development of sustainable cocoa agroforests

while conserving biodiversity. Such training centers would also include nurseries for the provision of local high-value agroforestry species considered important by local households and local market preferences.

**Author Contributions:** Conceptualization, C.K., A.K.Y. and A.R.A.; methodology, A.R.A., J.Z.G. and A.M.N.N.; software, A.M.N.N. and J.Z.G.; validation, A.R.A., T.d.K., C.K., A.K.Y. and A.M.N.N.; formal analysis, J.Z.G., A.R.A. and A.M.N.N.; investigation, T.d.K., A.K.Y., C.K. and A.R.A.; resources, C.K.; data curation, A.M.N.N., A.R.A. and A.M.N.N.; writing—original draft preparation, A.R.A.; writing—review and editing, A.R.A.; visualization, A.R.A., C.K., A.M.N.N., T.d.K. and J.Z.G.; supervision, C.K.; project administration, C.K. and T.d.K.; funding acquisition, C.K. All authors have read and agreed to the published version of the manuscript.

**Funding:** This work was jointly funded by Cémoi and Conseil du Café-Cacao, Côte d'Ivoire.

**Informed Consent Statement:** Written informed consent was obtained from the patient(s) to publish this paper.

**Data Availability Statement:** Data are available via the Dryad Digital Repository https://doi.org/10.5061/dryad.w9ghx3fn7 (Atangana et al., 2020).

**Acknowledgments:** This work was jointly supported by Cémoi and Conseil du Café-Cacao, Côte d'Ivoire. We thank Roger Leakey (International Tree Foundation) for his comments on a draft of this manuscript. We are also grateful to Landry Niava (Université Félix Houphouet-Boigny, Abidjan) and Lawani Adime (Ecole Nationale de Statistique Appliquée, Abidjan) for assistance in the sampling. Morou Yamiaman Siriki (ICRAF) assisted in the map drawing.

**Conflicts of Interest:** The authors declare no conflict of interest.

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
