# Peer review of "Rebuilding Tree Cover in Deforested Cocoa Landscapes in Côte d’Ivoire: Factors Affecting the Choice of Species Planted"

_forests, doi:10.3390/f12020198_

Round 1

Reviewer 1 Report

  1. The abstract reads moderately well. The sentence structure could be simplified at times, and punctuation improved. It could slightly shorter also. Consider simplifying the reporting of results; it's too much. Can you summarize key takeaways instead?
  2. Introduction - major grammar and sentence structure revisions required. The errors are distracting and make the content flow difficult to follow.
    1. An incomplete list of issues:
      1. Articles (a, the) often missing
      2. Passive voice frequently used
      3. Sentence structure can be simplified and reduced throughout
      4. Preposition use is sometimes incorrect
        1. For example, line 46 - change "of" to "for" 
    2. The objective is clear. 
  3. Materials and Methods
    1. Description of the question - It seems there should be more relevant details about survey question volume and type?
    2. Again. Difficulty to read, with run-on sentences, grammatical errors, and formatting mistakes. 
  4. Results
    1. The results seem thin. 
    2. I suggest combining the results and discussion sections. The discussion section is quite long and should be paired down, although it has ample good content.
    3. Since this paper is discussion heavy, I wonder if a different a different manuscript format (other than a research article) would be more appropriate? Possibly a perspective of plociy paper? 

Author Response

Rebuttal letter / forests-1043394

Subject: Manuscript ‘Rebuilding tree cover in deforested cocoa landscapes in Côte d’Ivoire: Factors affecting the choice of species planted’

Dear Editor,

Thank you very much for your letter of December , along with the comments from the two reviewers. We very much appreciate the comments and suggestions of the reviewers which helped improve the quality of the revised manuscript. As requested, the comments and queries have been addressed in the revised version of the manuscript.

The detailed point-wise reply of each comment is given below. Please find attached our revised version of the manuscript. We hope this new version is now satisfactory for publication in forests.

Point-wise reply

Editor:

I think the ms conveys interesting information but the authors must discuss their results in a more general and international context; otherwise, the ms is of too local scope. I noticed that the Abstract includes redundant information and must be improved.

Reply: The discussion was carried out in a more general and international context (lines 238-244, 254-259, 276-282 and 295-298). Redundant information in the Abstract section has been removed in the revised version. The Abstract has also been improved.

Reviewer #1

  1. The abstract reads moderately well. The sentence structure could be simplified at times, and punctuation improved. It could slightly shorter also. Consider simplifying the reporting of results; it's too much. Can you summarize key takeaways instead?

Reply: We submitted the manuscript to a thorough English editing; we also shortened the Abstract section. However, we believe that reporting the findings of this study in the Abstract section is the best option instead of the summarizing key takeaways, as many readers often have a look at the Abstract section only.

  1. Introduction - major grammar and sentence structure revisions required. The errors are distracting and make the content flow difficult to follow.
    1. An incomplete list of issues:
      1. Articles (a, the) often missing
      2. Passive voice frequently used
      3. Sentence structure can be simplified and reduced throughout
      4. Preposition use is sometimes incorrect
        1. For example, line 46 - change "of" to "for" 

Reply: The manuscript has gone a thorough English editing, and sentence structure has been simplified

    1. The objective is clear. 
  1. Materials and Methods
    1. Description of the question - It seems there should be more relevant details about survey question volume and type?

Reply: Details provided in the revised manuscript (lines 161-164); also, we provide questions as a supplementary material.

    1. Again. Difficulty to read, with run-on sentences, grammatical errors, and formatting mistakes. 

Reply: The manuscript was submitted to a thorough English editing prior to resubmission

  1. Results
    1. The results seem thin. 
    2. I suggest combining the results and discussion sections. The discussion section is quite long and should be paired down, although it has ample good content.

Reply: 1)We agree that the results are thin; this is because few factors effects were significant; 2) The last paragraph of the Discussion section has been removed

    1. Since this paper is discussion heavy, I wonder if a different a different manuscript format (other than a research article) would be more appropriate? Possibly a perspective of plociy paper? 

Reply: The manuscript reports the results of research work; we prefer to keep the current format, especially since adopting one of the suggested formats would not change the content of the manuscript

Reviewer #2

Cocoa agroforestry Cote d’Ivoire

The data collected for this manuscript can be the basis for an interesting article. But as it stands it needs major revision.

  • The standard in scientific writing is to have a topic sentence at the beginning of each sentence – then the remainder of the paragraph needs to stick to this subject…there is a bit too much “jumping around” here so needs tightening up
  • There are different bodies of literature we realize – “anglophone” versus “francophone” (and we hope soon “Africaphone”!) Nevertheless quite surprised that there is no mention of the works of Ruf and Schroth – simple searches on “biodiversity and cocoa and agroforestry” under their names will come up with at least one book and many articles specifically on the topics of this manuscript – so their perspectives should be considered.

Reply: The manuscript has been submitted to a thorough English editing; Ruf and Schroth made significant contributions to the understanding of relationships between agroforestry and biodiversity conservation (Ruf et al. 2004; Schroth and Harvey 2007; Casano et al. 2009; Ruth 2011; Schroth, Faria, Araujo et al. 2011; Schroth, da Motta, Hills et al. 2011); however, the studies date back to over 10 years ago; further, the present study is not solely based on biodiversity conservation through agroforestry

Many places where a correlation is seen as a cause – but we know “correlation is not causation” – there may be a connection but we do not know…So it should be clear in the text and captions that a correlation was found but a cause not necessarily…

Reply: Correlation shows the relationship between two variables, whether regressions allow us to see how one variable affects the other; the present study used regressions to see how factors that are reputed to influence agroforestry adoption influence tree species planting in cocoa farms

  • A scientific paper should not be an advertisement for a particular program or institution…or a plea for research funding. So perhaps should lighten the promotion of WAC – ICRAF. Not sure the solution – get agroforesters to recommend solutions – is necessarily to be recommended. I believe years ago ICRAF had program “diagnosis and design” which eventually became “design and disappear” because none of the systems devised were adopted. Much of the agroforestry literature either 1. Describes existing systems developed by local people or 2. Theoretical systems designed by scientists – spreadsheet exercises in which certain office-designed combinations appear profitable and beneficial but hardly are ever implemented in the real world, alley cropping with leguminous trees being the prime example. We scientists love it, the farmers out there trying to feed their families – not so much!

Reply: we do agree that a scientific paper should not be an advertisement for a particular program or institution; however, in Côte d’Ivoire, there is a vast program on rebuilding forest cover and ICRAF – WCA is particularly involved in the program; the program is in line with the Cocoa and Forest Initiative that seeks to reduce the negative impact of cocoa on forest; the program is also supported by the private sector

We are developing cocoa agroforestry basing heavily on the results of the present study; our systems are not ‘Theoretical’ nor ‘existing systems’; indeed, we take advantage on farmers’ preferences, local environment, positive ecological and economic interactions between components of the systems to co-develop (farmers have the option to agree or reject what has been developed prior to implementation of the system) cocoa agroforestry systems in Cote d’Ivoire

  • Perhaps explore a bit more the traditional forest-grown cocoa model – any hope of bringing it back? Mixing natural regeneration of forest with planted shade-tolerant cocoa? If the new model is planting single species of cocoa in sun and trees interfere with the yields what argument is there for introducing trees in to the system?

Reply: The ’traditional cocoa model’ in Côte d’Ivoire is full sun cocoa; natural regeneration of forests is one option taken by the Government of Cote d’Ivoire in highly invaded ‘Classified Forests”, along with cocoa agroforestry; the new model consists in planting several fast-growing species and bananas in new orchards one year before cocoa planting, so that bananas and other tree species will provide shade to young cocoa seedlings; the species should also have multiple uses, and their choice is made according to farmers’ preferences and objectives; we also aim at investigating the effects of cocoa companion species on soil fertility management, as our exploratory data shows that some species may have “liming effect” on acid soils of Cote d’Ivoire under cocoa

  • Value-chain challenges for tree species? Could say something more about issues involved in selling any wood that might come from trees planted? How long to sellable size? What are the impediments to small farmers commercializing their single trees? Policy-laws that favour large timber businessmen rather than individual landholders? Lack of capacity locally to process wood?

Reply: Value chain challenges are beyond the scope of this study; most of the tree species identified as potential ‘cocoa companion tree species’ in Cote d’Ivoire have multipurpose uses, on top of timber

  • Specific problems with some species Milicia – odum or iroko has serious problem with a gall-forming psyllid, very difficult to grow…..also Khaya spp. and other members of Meliaceae (mahogany and related) often ruined by shoot borers…any other specific information about trees that might or might not be compatible with cacao? Ceiba pentandra gets to be massive very quickly – will that be a good thing in a cacao plantation?

Reply: Yes, some species identified in the present study as potential cocoa companion species may host Cocoa swollen shoot diseases agents; we are currently screening the species for their ability to host cocoa diseases and pests, and the choice of the species to be included in cocoa farms will consider the results of the latter study

Reviewer 2 Report

Cocoa agroforestry Cote d’Ivoire

The data collected for this manuscript can be the basis for an interesting article. But as it stands it needs major revision.

  • The standard in scientific writing is to have a topic sentence at the beginning of each sentence – then the remainder of the paragraph needs to stick to this subject…there is a bit too much “jumping around” here so needs tightening up
  • There are different bodies of literature we realize – “anglophone” versus “francophone” (and we hope soon “Africaphone”!) Nevertheless quite surprised that there is no mention of the works of Ruf and Schroth – simple searches on “biodiversity and cocoa and agroforestry” under their names will come up with at least one book and many articles specifically on the topics of this manuscript – so their perspectives should be considered.
  • Many places where a correlation is seen as a cause – but we know “correlation is not causation” – there may be a connection but we do not know…So it should be clear in the text and captions that a correlation was found but a cause not necessarily…
  • A scientific paper should not be an advertisement for a particular program or institution…or a plea for research funding. So perhaps should lighten the promotion of WAC – ICRAF. Not sure the solution – get agroforesters to recommend solutions – is necessarily to be recommended. I believe years ago ICRAF had program “diagnosis and design” which eventually became “design and disappear” because none of the systems devised were adopted. Much of the agroforestry literature either 1. Describes existing systems developed by local people or 2. Theoretical systems designed by scientists – spreadsheet exercises in which certain office-designed combinations appear profitable and beneficial but hardly are ever implemented in the real world, alley cropping with leguminous trees being the prime example. We scientists love it, the farmers out there trying to feed their families – not so much!
  • Perhaps explore a bit more the traditional forest-grown cocoa model – any hope of bringing it back? Mixing natural regeneration of forest with planted shade-tolerant cocoa? If the new model is planting single species of cocoa in sun and trees interfere with the yields what argument is there for introducing trees in to the system?
  • Value-chain challenges for tree species? Could say something more about issues involved in selling any wood that might come from trees planted? How long to sellable size? What are the impediments to small farmers commercializing their single trees? Policy-laws that favour large timber businessmen rather than individual landholders? Lack of capacity locally to process wood?
  • Specific problems with some species Milicia – odum or iroko has serious problem with a gall-forming psyllid, very difficult to grow…..also Khaya spp. and other members of Meliaceae (mahogany and related) often ruined by shoot borers…any other specific information about trees that might or might not be compatible with cacao? Ceiba pentandra gets to be massive very quickly – will that be a good thing in a cacao plantation?

.

Author Response

(The authors gave the same response as above.)

Round 2

Reviewer 2 Report

References should not be rejected just because they are “old” – ten years – and in fact readers should be directed to the major publications on the topic, regardless of their age. Further it is not advised to state that something caused something else when all that is being presented is a correlation.

I do not know what the other referee (s) said – but there’s very little out there I think on this topic so a potential reader should be able to judge for her or himself the information presented here, taking everything “with a grain of salt”, take what’s good and forget the rest, as critical readers should do anyway.…So ok to publish in my opinion.